

# Identification of Organic Hydroperoxides and Peroxy Acids Using Atmospheric Pressure Chemical Ionization – Tandem Mass Spectrometry (APCI-MS/MS): Application to Secondary Organic Aerosol

Shouming Zhou[1], Jean C. Rivera-Rios[2], Frank N. Keutsch[2], Jonathan P.D. Abbatt[1]

[1]Department of Chemistry, University of Toronto, Canada; [2]Paulson School of Engineering and Applied Sciences and Department of Chemistry and Chemical Biology, Harvard University, U.S.A.

*Correspondence to*: Shouming Zhou (szhou@chem.utoronto.ca)

**Abstract.** Molecules with hydroperoxide functional groups are of extreme importance to both the atmospheric and biological chemistry fields. In this work, an analytical method is presented for the identification of organic hydroperoxides and peroxy acids (ROOH) by direct infusion of liquid samples into a positive-ion atmospheric pressure chemical ionization-tandem mass spectrometer ((+)-APCI-MS/MS). Under collisional dissociation conditions, a characteristic neutral loss of 51 Da (arising from loss of $H_2O_2+NH_3$) from ammonium adducts of the molecular ions ($[M+NH_4]^+$) is observed for ROOH standards (i.e. cumene hydroperoxide, isoprene-4-hydroxy-3-hydroperoxide (ISOPOOH), tert-butyl hydroperoxide, 2-butanone peroxide and peracetic acid), as well as the ROOH formed from the reactions of $H_2O_2$ with aldehydes (i.e. acetaldehyde, hexanal, glyoxal and methylglyoxal). This new ROOH detection method was applied to methanol extracts of secondary organic aerosol (SOA) material generated from ozonolysis of α-pinene, indicating a number of ROOH molecules in the SOA material. While the full scan mass spectrum of SOA demonstrates the presence of monomers (m/z=80-250), dimers (m/z=250-450) and trimers (m/z=450-600), the neutral loss scan shows that the ROOH products all have masses less than 300 Da, indicating that ROOH molecules may not contribute significantly to the SOA oligomeric content. We anticipate this method could also be applied to biological systems with considerable value.

## 1 Introduction

Organic hydroperoxides and peroxy acids (ROOH) are produced by the gas-phase oxidation of volatile organic compounds (VOCs) (Jackson et al., 1999; Lee et al., 2000; Atkinson and Arey, 2003), as well as in cloud and wet aerosols (Zhao et al., 2013; Lim and Turpin, 2015). Atmospheric oxidation of VOCs leads to secondary organic aerosol (SOA), an important fraction of the atmospheric aerosol burden. Both modeling and experimental studies indicate that organic peroxides (i.e. ROOR and ROOH species) are major components of SOA (Jenkin. 2004; Bonn et al., 2004; Docherty, et al., 2005). In recent years, ROOH in particular have been proposed to be involved in high molecular weight products leading to SOA formation (Krapf et al., 2016; Kristensen et al., 2016; Sakamoto et al., 2017). Organic peroxides are also widely used industrially as radical initiators, bleaching and disinfecting agents, and reactive intermediates in the polymer, food, and pharmaceutical



industries (Odian, 2004; Moll et al., 1979; Reile et al., 2011). In biological systems, ROOH are formed from the reactions of radicals and singlet oxygen with amino acids, peptides, and proteins (Gebicki and Gebicki, 1993; Wright et al., 2002; Agon wt al., 2006; Morgan et al., 2008). Classified as one component of reactive oxygen species (ROS), ROOH are hazardous, irritating to skin, eyes, and mucous membranes. They also cause progressive oxidative damage, cell death, and even cancer

(Liou and Storz, 2010).

Despite their importance in both atmospheric and biological chemistry, the identification of specific ROOH molecules in a complex mixture remains analytically challenging. There are two reasons for this: i) unavailability of the ROOH standards because of their thermally unstable nature (Bach et al., 1996), and ii) the lack of appropriate analytical techniques. So far, the analysis of ROOH in the condensed phase has mainly been done by means of chemical assays, such as the iodometric

(Docherty et al., 2005; Banerjee and Budke, 1964), triphenylphosphine (Nakamura and Maeda, 1996), ferrous oxidation-xylenol orange (Wasylaschuk et al., 2007), and horseradish peroxidase approaches (Walker et al., 2006; Hong et al., 2008). These techniques react ROOH with reducing agents followed by analysis of the reaction products. ROOH have also been analyzed by HPLC analysis followed by post-column chemical derivatization method (Valverde-Canossa et al., 2005; Francois et al., 2005; Hasson et al., 2001).

The disadvantage of the assay techniques is that they measure total peroxide content (some combination of organic peroxides, organic hydroperoxides, and hydrogen peroxide) and are not able to identify specific molecules. There are limited studies on direct analysis of ROOH in the literature. Using ESI-MS, Hui et al. identified ROOH formed from the oxidation of cholesteryl ester, reporting ammonium and sodium adducts of the ROOH molecular ions under positive ion mode and acetate adducts of ROOH molecular ions under negative ion mode (Hui et al., 2012a; 2012b). Reinnig et al. (2009) identified

ROOH using on-line ion trap mass spectrometry, proposing that neutral loss of 34 Da (loss of $H_2O_2$) from the protonated molecular ions is a characteristic fragmentation route for ROOH. Using this technique the authors identified 3 ROOH molecules from SOA generated from reaction of α-pinene and ozone (Reinnig et al., 2009).

Recently, using high-resolution mass spectrometry, a number of studies proposed ROOH detection arising from atmospheric oxidation of biogenic organics (Zhang et al., 2017; Riva et al., 2017). However, due to the fact that the high-resolution mass

spectrometry can only provide elemental composition of the molecules, the identification of ROOH in the reaction systems remains speculative.

In the present work, a positive-ion atmospheric pressure chemical ionization-tandem mass spectrometer ((+)-APCI-MS/MS) is applied to identify specific ROOH molecules. The analytical method is developed by using ROOH commercial standards and ROOH molecules that are generated from the reactions of aldehydes with $H_2O_2$. The method is applied to SOA formed

from ozonolysis of α-pinene and a number of ROOH molecules are identified. The goal of this work is to provide an analytical technique that can widely be applied in not only the atmospheric chemistry field, but also other settings.

## 2 Experimental Section

### 2.1 Chemicals and Reagents.



Cumene hydroperoxide (80%), tert-butyl hydroperoxide (35%), 2-butanone peroxide (35%), peracetic acid (32%), di-tert-butyl peroxide (98%), benzoyl peroxide (≥98%), di(dodecanoyl) peroxide (97%), 2-nonenal (97%), meso-erythritol (≥99%), cis-pinonic acid (98%), formaldehyde (37%), acetaldehyde (>99%), hexanal (98%), methyl glyoxal (40%), glyoxal (40%), hydrogen peroxide ($H_2O_2$, 30%), α-pinene (≥99%) and ammonium acetate (≥99.99%) are all purchased from Sigma-Aldrich

(Canada). Isoprene-4-hydrox-3-hydroperoxide (ISOPOOH) is synthesized according to literature (Rivera-Rios et al., 2014). Methanol (MeOH, LC-MS grade) is purchased from VWR, Canada. All the chemicals and reagents are used as received.

**2.2 Sample Preparation and Reactions of Aldehydes with $H_2O_2$.**

Stock solutions for ROOH standards (i.e. cumene hydroperoxide, ISOPOOH, tert-butyl hydroperoxide, 2-butanone peroxide and peracetic acid), other peroxides (di-tert-butyl peroxide, benzoyl peroxide and di(dodecanoyl) peroxide), 2-nonenal,

meso-erythritol and cis-pinonic acid are prepared by dissolution of the substances in MeOH (2.0-10 mM). 100 μL stock solutions are further diluted in MeOH to a final volume of 1 mL for mass spectrometry analysis. In some cases, ammonium acetate (AA) is added (~5 mM) to enhance the signal for the ammonium adducts of the molecular ions in the mass spectra.

Reactions of the selected aldehydes (i.e. formaldehyde, acetaldehyde, hexanal, glyoxal and methyl glyoxal) with $H_2O_2$ are performed by mixing the reactants in MeOH (~3 mM and ~13 mM for aldehdyes and $H_2O_2$, respectively) at room

temperature (295±3K) for ~10 minutes. AA is added (~5 mM) before the samples are analyzed. Sample blanks (i.e. aldehydes+AA and $H_2O_2$+AA) are prepared and analyzed in the same manner as the reaction mixtures.

**2.3 Secondary Organic Aerosol (SOA) Generation and Collection.**

SOA is generated in a steady-state manner in a 1 $m^3$ Teflon (FEP) chamber from gas-phase reaction of α-pinene with ozone (Aljawhary et al., 2013). α-pinene is introduced into the chamber by passing a ~2 mL $min^{-1}$ nitrogen through a bubbler

containing α-pinene, that is chilled to -20 °C and mixed with 5 L $min^{-1}$ purified air. Ozone is generated by passing ~10 L $min^{-1}$ purified air through a mercury lamp. The final concentration of ozone is monitored to be ~7.4×$10^{12}$ molecules $cm^{-3}$ by a UV photometric ozone analyzer (Thermo Model 49i) and the α-pinene concentration is estimated to be ~3.7×$10^{12}$ molecules $cm^{-3}$.

SOA generation is confirmed by a scanning mobility particle sizer (SMPS, Model 3034) and collected on quartz fiber filters

(47 mm diameter) for 72 hours at a flow of ~15 L $min^{-1}$. The filters are preheated at 500 °C for 24 hours to remove organic impurities before SOA collection. After SOA collection, the filter is extracted with 10 mL MeOH and the extract is immediately analyzed by the mass spectrometer.

**2.4 Atmospheric Pressure Chemical Ionization – Tandem Mass Spectrometer (APCI-MS/MS).**

A unit resolution APCI-MS/MS instrument (Thermo TSQ Endura) is operated in positive ion mode with direct infusion of

the samples into the mass spectrometer. The sample is injected at a flow rate of 10 μL $min^{-1}$ using a syringe pump (Chemyx Inc. U.S.A., Model Fusion 101) to the APCI source through polyether ether ketone (PEEK) tubing. The spray voltage is set at +2500 V; vaporizer temperature and ion transfer tubing temperature are set at 200 °C. Sheath gas, auxiliary gas and sweep gas flows are set (arbitrary units) at 5, 2 and 0, respectively. The mass spectrometer is a triple quadrupole that is calibrated with polytyrosine. Mass spectra are obtained under either full scan mode or selected ion monitoring (SIM) mode.



In full scan mode, RF lens voltages are the default from the optimization in the mass calibration. The SIM scan is achieved by isolating and monitoring a range of masses with maximum mass range of 50 Da. The RF lens voltages in SIM mode can be manually varied and are optimized to maximize the intensities of the ammonium adducts of the molecular ions.

Ion fragmentation in the MS/MS is accomplished by application of electrical potentials (2-10 V) and collision gas (Argon,

0.5 mTorr) in the collision-induced dissociation (CID) cell. Mass spectra from product scans are obtained by transmitting ions of a specific m/z through the first quadrupole (precursor ions), fragmenting in the CID cell, and monitoring the resulting fragment ions (product ions) by the third quadrupole.

In the neutral loss (NL) scan mode, the first and third quadrupoles are scanned at the same rate over mass ranges of the same width, i.e. the third quadrupole transmits ions at a fixed mass-to-charge ratio lower than the first quadrupole. In the NL scan,

the CID voltage and argon pressure are set at 2-10 V and 0.5 mTorr, respectively. The LOD of the analytical method is established by direct injection of the ROOH standard solution in MeOH with 2-10 mM ammonium acetate into the APCI source and the tandem mass spectrometer being operated under NL scan mode. Three ROOH standards, namely 2-butanone peroxide, tert-butyl hydroperoxide, and cumene hydroperoxide, are analysed. As will be seen in the next section that a neutral loss of 51 Da from the ammoniated molecular ion ($[M+NH_4]^+$) of the ROOH is characteristic for ROOH molecules,

the calibration is performed by operating with a loss of 51 Da in NL scan. The ROOH concentrations are selected when the $[M+NH_4]^+$ ions are well above the noise and clearly seen in the average mass spectra. The LOD is reported as 3 times the S/N where noise is estimated at mass-to-charge ratios different from the mass-to-charge ratio for $[M+NH_4]^+$.

### 3 Results and Discussion

**3.1 APCI Mass Spectra of ROOH Standards**

Fig. 1 presents examples of the direct infusion (+)-APCI mass spectra for: (A) cumene hydroperoxide (cumene HP), and (B) 4,3-ISOPOOH, with and without addition of AA. The chemical structures and molecular weights (MW) of the ROOH analyzed are given in Fig. SI1. Note that we do not attempt to interpret the mass spectra of ROOH and other standards obtained under full scan and SIM modes due to the presence of stabilizers and other impurities in the standard samples that

make the mass spectra complex. Instead, we focus on the protonated and ammoniated molecular ions of the ROOH molecules.

It can be seen from Fig. 1 that, although cumene HP demonstrates the proton adduct of the molecular ion at m/z=153 ($[M+H]^+$ (top panel of Fig. 1 (A)), addition of AA results in the ammonium adduct at m/z=170 ($[M+NH_4]^+$ (bottom panel of Fig. 1 (A)). For ISOPOOH, the molecular ion is not seen in the full scan mass spectrum (top panel of Fig. 1 (B)) and the

dehydrated molecular ion at m/z=101 ($[M+H-H_2O]^+$) is clearly observed instead (top panel of Fig. 1 (B)). This is consistent with previous studies claiming that hydroperoxy group (-OOH) is not a favorable protonation or deprotonation site with ESI or APCI (Reinnig et al., 2008; Rondeau et al., 2003; Nilsson et al., 2008). The addition of AA again leads to significant production of the ammonium adduct of the molecular ion at m/z 136 (bottom panel of Fig. 1 (B)). Similar effects of AA on the APCI mass spectra are observed for other ROOH species.



In comparison, a number of other common molecules in atmospheric samples, e.g. organic peroxides (ROOR), carbonyls, alcohols and carboxylic acids, were also analyzed. The chemical structures of the other oxygenated organics are given in Fig. SI2. Similar to the ROOH samples, the ammonium adducts of the molecular ions ($[M+NH_4]^+$) are obtained for all the substances. The only difference is that in some cases, e.g. benzoyl peroxide, the ammonium adduct of the molecular ions can

be clearly seen without addition of AA, via trace levels of ammonia present in the water or air.

**3.2 Product Spectra of $[M+NH_4]^+$ for ROOH and Other Organics**

Fig. 2 gives the CID fragment patterns (i.e. product spectra) for ammonium adducts of the molecular ions ($[M+NH_4]^+$) of: (A) cumene HP, and (B) ISOPOOH. Two types of fragmentation, i.e. loss of 35 and 51 Da from the respective $[M+NH_4]^+$ ions are observed in both cumene HP and ISOPOOH. The loss of 35 Da, corresponding to $[-H_2O-NH_3]$ (Fig. SI3) is also

observed in meso-erythritol and pinonic acid (Fig. SI4). Neutral loss of 51 Da, corresponding to $[-H_2O_2-NH_3]$ (Fig. SI3) is only observed from fragmentation of the $[M+NH_4]^+$ ions of ROOH (Fig. 2 and SI4). It is known that the loss of the $H_2O$ molecule from protonated molecular ions ($[M+H]^+$) is typical for epoxides, alcohols and carboxylic acids (Holcapek et al., 2010). But the loss of 35 Da from $[M+NH_4]^+$ is not characteristic for ROOH. Instead, the loss of 51 Da from $[M+NH_4]^+$ is only observed in ROOH standards, including peracetic acid (Fig. SI5). Hence we propose that the neutral loss of 51 Da from

$[M+NH_4]^+$ is characteristic for ROOH molecules. This is consistent with previous work that suggested that neutral loss of $H_2O_2$ from $[M+H]^+$ is also characteristic for ROOH species (Reinnig et al., 2009).

**3.3 ROOH Formation from the Reactions of Aldehydes and $H_2O_2$**

Using [1]H NMR spectroscopy, it has been shown that ROOH species form from the reactions of aldehydes with $H_2O_2$ (Zhao et al., 2013). As shown in Fig. 3 the reaction proceeds via reversible nucleophilic addition of $H_2O_2$ to the carbonyl group in

aldehydes, leading to alpha-hydroxyhydroperoxides (HHP). The addition of $H_2O_2$ to methylglyoxal (MGL, MW=72) gives rise to methylglyoxal hydroxyhydroperoxide (MGL HHP, MW=106) (Fig. 3 (A)) whose ammonium adduct is seen at m/z 124 in Fig. 4 (A). The mass spectrum of the reaction mixture of glyoxal and $H_2O_2$ suggests that rather than a direct addition to glyoxal, $H_2O_2$ is instead added to the glyoxal geminal diol formed by the hydrolysis of glyoxal (Fig. 3 (B)), producing $[M+NH_4]^+$ at m/z 128 (Fig. SI6 (A)). The $[M+NH_4]^+$ ROOH peaks are also observed in the reactions of other aldehydes with

$H_2O_2$ (data not shown).

The fragmentation spectra of the $[M+NH_4]^+$ of the ROOH from methylglyoxal and glyoxal are given in Fig. 4 (B) and Fig. SI6 (B), respectively. It is clear that the neutral loss of 51 Da is again observed in the fragments of the ROOH molecules. The ROOH products from all the other aldehyde reactions with $H_2O_2$ also show loss of 51 Da in MS/MS mode. Overall, we conclude that the neutral loss of 51 Da from $[M+NH_4]^+$ fragmentation can be used to identify ROOH molecules.

**3.4 Identification of ROOH in SOA Material**

Fig. 5 presents mass spectra of the methanol extract of secondary organic aerosol (SOA) generated from ozonolysis of α-pinene under dry conditions (RH<5%). There are three features of note. First, the well-characterized products from this reaction, such as norpinone aldehyde (MW=154), terpenylic acid (MW=172) and cis-pinic acid (MW=186) (Jenkin et al., 2000; Larsen et al., 2001; Claeys et al., 2009) are present as protonated molecular ions ($[M+H]^+$) in the full scan mode (Fig.



5 (A)). Second, the full scan mass spectrum (Fig. 5 (A)) shows the presence of monomer (m/z=80-250), dimer (m/z=250-450) and trimer (m/z=450-600) species, as has been reported in previous work (Venkatachari and Hopke, 2008). Third, and perhaps most importantly, the 51 Da neutral loss scan (Fig. 5 (B)) indicates that all the ROOH species have masses <300 Da (Fig. 5(B)).

To confirm the observation of ROOH obtained with the neutral loss scan, CID fragmentation spectra of a few intense peaks at m/z 190, 202, 206, 218 and 220 in Fig. 5 (B) are analyzed. All the fragments of these products show loss of 51 Da. An example of the fragment pattern for m/z 206 is given in Fig. SI7.

The ROOH products in SOA are tentatively identified and listed in Table 1 and mechanisms for the formation of these species are presented in Fig. SI8. Note that plausible mechanisms to form most of the ROOH species identified in Fig. 5 (B)

can be written, as indicated by boxed structures in Fig. SI8.

Several mechanisms for oligomer product formation in SOA arising from VOC oxidation have been proposed: i) self- and cross-reactions of the peroxy radicals ($RO_2$) (Zhang et al., 2016), ii) reaction of ozonolysis products in the condensed-phase, such as aldol condensation, esterification, hemiacetal and peroxyhemiacetal formation (Ziemann, 2003; Tolocka et al., 2004; Kristensen et al., 2014; Docherty et al., 2005; Muller et al., 2009; Yasmeen et al., 2010; Hall and Johnston, 2012; Witkowski

and Gierczak, 2012; DePalma et al., 2013; Lim and Turpin, 2015), iii) dimer cluster formation from carboxylic acids (Hoffmann et al., 1998; Tobias and Ziemann, 2000; Claeys et al., 2009; Camredon et al., 2010; DePalma et al., 2013), iv) reactions of Criegee intermediates (CIs) with VOCs oxidation products (Bonn et al., 2002; Lee and Kamens, 2005; Tolocka et al., 2006; Heaton et al., 2007; Witkowski and Gierczak, 2012; Kristensen et al., 2016; Wang et al., 2016), and vi) reactions of $RO_2$ radicals with Cis (Sadezky et al., 2008; Zhao et al., 2015). Among them, the reactions of CIs with protic substances

(water, alcohols, acids and hydroperoxides) can form ROOH. However, the nature of the ROOH products observed in SOA material suggests that these reactions do not take place to a significant extent given that ROOH do not contribute significantly to the dimer and trimer SOA signals (Fig. 5(B)). The similar mass patterns for the SOA and ROOH obtained under dry (RH<5%) and humid (RH=50%) conditions support this conclusion (Fig. SI9). Of course, we can not rule out that the oligomeric ROOH may not be sensitive with the analytical method used. As well, while the ROOH formation

mechanisms are proposed via gas-phase ozonolysis of α-pinene (Fig. SI8), the ROOH observed in the SOA could also potentially be arising from the decomposition of peroxyhemiacetals during methanol extraction.

The neutral loss scan mode, the LOD were measured to be 0.2, 0.3 and 20 mM for 2-butanone peroxide, cumene hydroperoxide and tert-butyl hydroperoxide, respectively. The LODs obtained by this method are a rough estimation and will vary dependent on a number of parameters: ionization voltage, sample injection flow, gas flow, CID gas pressure, and

CID voltages, etc. More importantly, if a specific ROOH is to be analysed, then multiple reaction monitoring (MRM) mode would be applied and the LOD of the ROOH would be substantially reduced.

**4 Conclusions**





Organic hydroperoxides are molecules of crucial importance to atmospheric chemistry, arising under VOC oxidation schemes that proceed under low NOx conditions. Indeed, as NOx levels continue to drop throughout many parts of the atmosphere through emission control measures, it is expected that these species will become even more prevalent. Furthermore, many of these ROOH molecules are known to constitute an important component of secondary organic aerosol

material. Once in atmospheric particles, ROOH can participate in condensed-phase reactions including nucleophilic processes and photolysis, and they are likely harmful when deposited into lung fluid.

A major complication in the study of ROOH chemistry has been the lack of detection techniques that are able to identify different ROOH species. In this work we present an off-line method for the identification of aqueous phase ROOH molecules which involves first ionization by adduct formation with ammonium ions, and then collision induced dissociation

by a unique fragmentation pathway involving the simultaneous loss of both $NH_3$ and $H_2O_2$. Although only a fraction of the total ion fragmentation pathway involves 51 Da neutral loss, the specificity of the tandem mass spectrometry approach will yield low detection limits.

We illustrate the utility of this analytical method by demonstrating that a set of ROOH molecules is present in α-pinene ozonolysis SOA, all arising from known oxidation mechanisms. Perhaps most interestingly, ROOH species were not

observed to be present in the oligomeric fraction of α-pinene ozonolysis SOA, indicating that reactions such as Criegee radicals reacting with protic substances are not a source of such dimeric and trimeric molecules.[5] In this manner, we believe that this new analytical approach could be used widely to decipher the prevalence of ROOH molecules in different forms of SOA. Given the specificity of the method, it could also be used to monitor the kinetics of condensed phase reactions of individual ROOH molecules.

Although this work focused on the use of this new analytical method to analyze for atmospheric ROOH molecules, it could equally well be applied to the detection of ROOH molecules in other systems, especially biological ones.

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

**Acknowledgements.**

This work was supported by the Alfred P. Sloan Foundation and NSERC. F.N.K. and J.C.R. would also like to acknowledge the support of the National Science Foundation (AGS 1628491, 1628530, 1247421, and 1321987).






**Figure 1.** (+)-APCI mass spectra of ROOH with and without addition of ammonium acetate (AA) for (A) cumene HP and

5  (B) ISOPOOH.



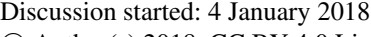

**Figure 2.** Product spectra of $[M+NH_4^+]^+$ for (A) cumene HP and (B) ISOPOOH.



**(A)**



methylglyoxal
(MGL), MW=72

MGL HHP, MW=106

**(B)**

glyoxal
(GXL), MW=58

geminal diol

GXL HHP, MW=110

**Figure 3.** ROOH formation from reactions of $H_2O_2$ with (A) methylglyoxal, and (B) glyoxal.





**Figure 4.** (A) Mass spectrum of reaction of methylglyoxal with $H_2O_2$ in the presence of AA; (B) Product mass spectrum of m/z 124 from (A).



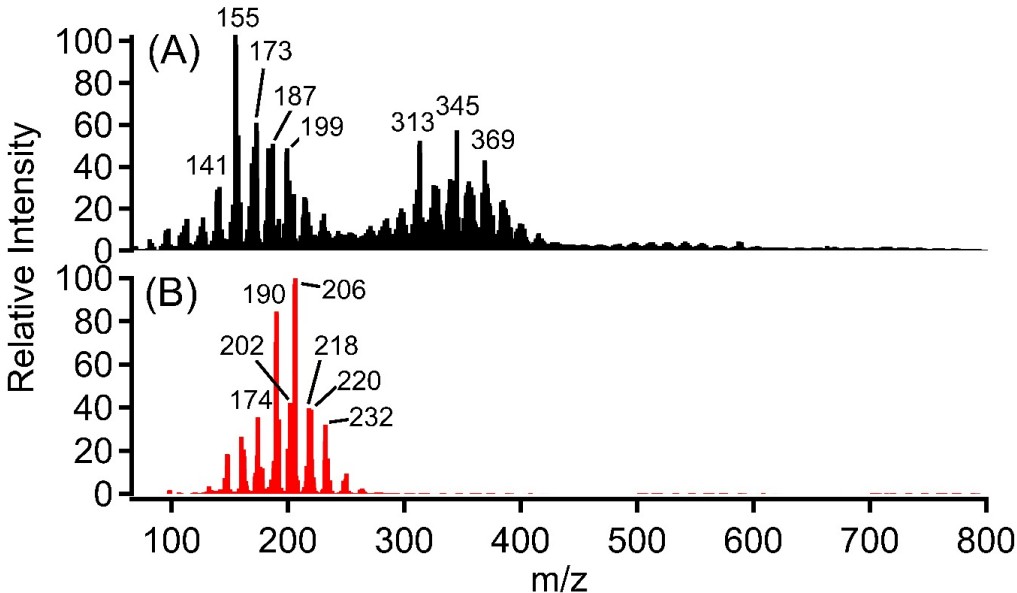

**Figure 5.** Mass spectra of SOA from ozone reaction with α-pinene under dry conditions (RH<5%) obtained with (A) full scan, and (B) neutral loss scan of 51 Da.





**Table 1. Possible Identities of the ROOH in α-pinene SOA**

| m/z ([M+NH₄⁺]⁺) | Molecular Weight (MW) | Chemical Structure |
|---|---|---|
| 174 | 156 | |
| 190 | 172 | |
| 202 | 184 | |
| 206 | 188 | |
| 218 | 200 | |
| 220 | 202 | |
| 232 | 214 | |