# Peer review of "Identification of Organic Hydroperoxides and Peroxy Acids Using Atmospheric Pressure Chemical Ionization — Tandem Mass Spectrometry (APCI-MS/MS): Application to Secondary Organic Aerosol"

_Atmospheric Measurement Techniques, 2017_

## Referee Comment (RC1) · Anonymous Referee #3 · 2 Feb 2018

Zhou et al. propose a new method to detect the hydroperoxides and peroxy acids in complex matrices such as organic aerosol. This analytical method is based on atmospheric pressure chemical ionization coupled with a mass spectrometer. While this study presents a promising method, additional discussions/information are needed before the publication.

General comments: What is the purity of the standards synthesized? It is important to

know the presence of potential artifacts to better understand the MS data. NMR data would be appreciated.

Page 3. Lines 31-32: As the authors mentioned the hydroperoxides are thermally unstable, so did they try to optimize the temperature of the vaporizer? Does it have an effect?

Page 4. Lines 23-26: it is not clear why the authors decided to not look at the MS into more details. I can understand that in the MS without the addition of AA it is not necessarily worth it but in the presence of AA I do think it is important to have a closer look to check if other fragmentations (specific?) occurred. So the authors should dig into the MS to look at the presence of other specific fragments or clusters.

Page 5. Line 5 Please provide the product spectra and MS/MS spectra for all the standards tested. It is important to have such information.

Page 5. Lines 7-16: The authors need to provide more information on the optimization of their method. It seems that the method leads to important fragmentations of the molecular ions and the fingerprint fragment ions have generally a low intensity. Have they tried to maximize the neutral loss of 51? If the authors haven't tried to optimize this aspect, what was the reason?

Page 5. Line 25. Similar to my previous comment, the authors need to present as much data as they can in order to demonstrate the robustness of the technique. In addition, some of the MS are quite noisy and lead to the question of the sensitivity of this analytical method. While it is promising and it seems very selective for the analyses of hydroperoxides, the results presented here indicate that it is not a sensitive method. What is the SOA mass collected from the chamber experiments? The LOD reported page 6 are very bad ($\sim$ mg L-1) compare to the current standards in the characterization of organic aerosol ($\sim$ ng L-1). The authors should discuss this issue especially in the perspective of the chemical characterization of hydroperoxides within environmental matrices (e.g. SOA) and the instability of such products.

Page 7. line 10-12 The authors claim that MS/MS will yield low detection limit and allow the identification of the hydroperoxides & peroxy acids. Based on the MS/MS spectra presented and the concentrations used in this study I don't think the authors can claim to propose a method with "a low detection limit". In addition, the authors need to report in the captions the concentrations of the solution and/or the mass of organics needed to generate the MS data.

Page 7. Lines 14-16 The authors should propose similar Figure as Figure 5 for the SOA generated under humid conditions. The authors have to be prudent in their conclusions and keep in mind the high reactivity of oligomeric products formed from the reaction of Criegee radicals with protic substances (e.g. Riva et al., 2017 Atmos Environ). Indeed, such species can have been degraded during the 72 hours of sampling and/or throughout the analytical protocol. Did the authors estimate the degradation of the hydroperoxides/peracid?

Technical comments: $m/z$ should be in italic.

Page 6. Proper references for the formation of dimers: Crounse et al 2013 and Ehn et al. 2014.

---

## Referee Comment (RC2) · Anonymous Referee #2 · 7 Feb 2018

Summary

This AMTD article a new analytical method that the author developed to identify organic hydroperoxides and peroxy acids. One of the unique properties of this technique is that it can direct analyze liquid samples using a positive-ion CI-MS/MS under atmospheric pressure. This study used pure standards first to verify the technique, and then measured ROOH in alpha-pinene ozonolysis SOA. Overall, the manuscript is sound

and after addressing the following issues, it is suitable to be published on AMT.

Comments

Page 3, section 2.3: the author described the method used to collect SOA filters. There is no SOA mass concentration nor the mass of SOA collected on the filter reported. Since the author mentioned "SOA generation is confirmed by a SMPS", then number-diameter distribution of the SOA should be put in the SI. The author should also report the SOA mass concentration and estimate the mass of SOA collected on the filter so it will give readers a reference point about how much SOA mass was analyzed.

The author concluded in page 6, lines 20-23 that ROOH do not contribute significantly to the dimer and trimer SOA signals. Then in page 3, line 32, the author described :" vaporizer temperature and ion transfer tubing temperature are set at 200 °C". Previous studies have shown that oligomers from alpha-pinene SOA can thermally decompose at 100 °C (Hall and Johnston 2012, Williams, Zhang et al. 2016). Because the author heated it to 200 °C, would be possible that a lot of oligomers are decomposed to form small ROOH molecules when passing through the vaporizer and the ion transfer tubing, leading to a bias of the current results?

References

Hall, W. A. and M. V. Johnston (2012). "The Thermal-Stability of Oligomers in Alpha-Pinene Secondary Organic Aerosol." Aerosol Sci. Technol. 46(9): 983-989. Williams, B. J., et al. (2016). "Organic and Inorganic Decomposition Products from the Thermal Desorption of Atmospheric Particles." Atmos. Meas. Tech. 9(4): 1569-1586.

---

## Referee Comment (RC3) · Anonymous Referee #1 · 9 Feb 2018

General Comments

In this manuscript the authors describe a new method for detecting hydroperoxides and peroxy acids in organic aerosol collected in filter samples. The method involves analysis of the extract following addition of ammonium acetate (when necessary) by atmospheric pressure ionization tandem mass spectrometry. The ammonium adducts $(M + NH_4)^+$ of the hydroperoxides and peroxy acids are observed in the mass spectrum and characteristically show loss of mass 51 due to H2O2 + NH3. The method has been evaluated for a large suite of compounds relevant to the atmosphere, showing similar behavior for all, and it has also been shown that the compounds without hydroperoxide or peroxy acid groups do not show the mass 51 loss. The methods were employed to analyze SOA formed from the reaction of a-pinene with O3, and only monomer hydroperoxides and peroxy acids were identified (no dimers or larger oligomers). The method is an important advance for atmospheric chemistry, since organic peroxides are a class of compounds whose importance is growing rapidly because of decreases in NOx and the recognition of the importance of atmospheric autoxidation. The manuscript is clear and concisely written, and I highly recommend it be published after the following minor comments are addressed.

Specific Comments

1. I wonder if the absence of detected peroxide dimers or other oligomers in the SOA could be due to decomposition that occurred prior to analysis. Krapf et al., Chem., 2016 measured a lifetime for organic peroxides in SOA formed from a-pinene ozonolysis of a few hours. And the organic chemistry literature indicates that peroxide decomposition pathways generally involve Baeyer-Villager reactions in which hydroperoxides or peroxyacids react with aldehydes or ketones to first form peroxyhemiacetals or acylperoxyhemiacetals (these would be observed as dimers), which then decompose to acids, esters, and alcohols. So it may be that once a dimer forms, it decomposes too soon to be observed here. This would be exacerbated by the long SOA sampling time of 72 hours. Perhaps some would be detected if the sampling time was shortened.

2. I am not sure of the necessity to speculate about mechanisms for formation of peroxides in Figure SI8. There is still very little known about these pathways and there are many proposed steps to getting to products, so to me it would be perfectly appropriate (and the authors would be on more solid ground) to just speculate on the structures of the products based on the observed mass spectrometry results. As a few examples of this issue, I believe it is well established now from structure activity calculations

(Vereecken & Peeters, PCCP, 2009, 2010) that radicals with RC(O)O structures immediately lose CO2. It is also not clear how an internal CO is lost in one of the steps shown, and I am not sure that the 184 compound is stable. I think it is just an excited intermediate that immediately loses OH. If the authors prefer to keep all this, then it would probably be useful to provide some references (perhaps Orlando and Tyndall, Chem. Soc. Rev. 2012 and elsewhere) to support the various proposed pathways.

Technical Comments

None.
* * *

---

## Author Comment (AC1) · 16 Apr 2018

General Comments:

In this manuscript the authors describe a new method for detecting hydroperoxides and peroxy acids in organic aerosol collected in filter samples. The method involves analysis of the extract following addition of ammonium acetate (when necessary) by atmospheric pressure ionization tandem mass spectrometry. The ammonium adducts

(M+NH4)+ of the hydroperoxides and peroxy acids are observed in the mass spectrum and characteristically show loss of mass 51 due to H2O2+NH3. The method has been evaluated for a large suite of compounds relevant to the atmosphere, showing similar behavior for all, and it has also been shown that the compounds without hydroperoxide or peroxy acid groups do not show the mass 51 loss. The methods were employed to analyze SOA formed from the reaction of a-pinene with O3, and only monomer hydroperoxides and peroxy acids were identified (no dimers or larger oligomers). The method is an important advance for atmospheric chemistry, since organic peroxides are a class of compounds whose importance is growing rapidly because of decreases in NOx and the recognition of the importance of atmospheric autoxidation. The manuscript is clear and concisely written, and I highly recommend it be published after the following minor comments are addressed.

Response: We thank the reviewer for the positive comments on our work.

Specific Comments:

I wonder if the absence of detected peroxide dimers or other oligomers in the SOA could be due to decomposition that occurred prior to analysis. Krapf et al., Chem., 2016 measured a lifetime for organic peroxides in SOA formed from a-pinene ozonolysis of a few hours. And the organic chemistry literature indicates that peroxide decomposition pathways generally involve Baeyer-Villager reactions in which hydroperoxides or peroxyacids react with aldehydes or ketones to first form peroxyhemiacetals or acylperoxyhemiacetals (these would be observed as dimers), which then decompose to acids, esters, and alcohols. So it may be that once a dimer forms, it decomposes too soon to be observed here. This would be exacerbated by the long SOA sampling time of 72 hours. Perhaps some would be detected if the sampling time was shortened.

Response: Thanks for the suggestion. We now make clear in the paper the possibility that oligomeric ROOH species may not be stable, either during collection or at the higher temperatures used in the vaporizer lines of the mass spectrometer source

(page 6 line32-33). That being said, we did observe dimers and trimers in a-pinene ozonolysis products (see Figure 5A) and mentioned the possible formation mechanism as the reactions of hydroperoxides or peroxyacids with aldehydes or ketones in the manuscript (page 6 line 21-24). However, because the peroxyhemiacetals or acylperoxyhemiacetals are not ROOH type of products, they do not show the characteristic neutral loss of 51 Da from their ammoniated molecular ions.

I am not sure of the necessity to speculate about mechanisms for formation of peroxides in Figure SI8. There is still very little known about these pathways and there are many proposed steps to getting to products, so to me it would be perfectly appropriate (and the authors would be on more solid ground) to just speculate on the structures of the products based on the observed mass spectrometry results. As a few examples of this issue, I believe it is well established now from structure activity calculations (Vereecken & Peeters, PCCP, 2009, 2010) that radicals with RC(O)O structures immediately lose $CO_2$. It is also not clear how an internal CO is lost in one of the steps shown, and I am not sure that the 184 compound is stable. I think it is just an excited intermediate that immediately loses OH. If the authors prefer to keep all this, then it would probably be useful to provide some references (perhaps Orlando and Tyndall, Chem. Soc. Rev. 2012 and elsewhere) to support the various proposed pathways.

Response: We agree that the mechanisms proposed in Figure SI8 were speculative and they have been removed from the revised manuscript. We added the following sentences in the revised manuscript (page 6 line 8-9): "The ROOH products in SOA are tentatively identified and listed in Table 1. It should be noted that the chemical structures of these products are only proposed from their molecular ions and therefore remain speculative."

Technical Comments: None.

---

## Author Comment (AC2) · 16 Apr 2018

Summary:

This AMTD article a new analytical method that the author developed to identify organic hydroperoxides and peroxy acids. One of the unique properties of this technique is that it can direct analyze liquid samples using a positive-ion CI-MS/MS under atmospheric pressure. This study used pure standards first to verify the technique, and then

measured ROOH in alpha-pinene ozonolysis SOA. Overall, the manuscript is sound and after addressing the following issues, it is suitable to be published on AMT.

Response: We thank the reviewer for the positive comments on our work.

Comments:

Page 3, section 2.3: the author described the method used to collect SOA filters. There is no SOA mass concentration nor the mass of SOA collected on the filter reported. Since the author mentioned "SOA generation is confirmed by a SMPS", then number-diameter distribution of the SOA should be put in the SI. The author should also report the SOA mass concentration and estimate the mass of SOA collected on the filter so it will give readers a reference point about how much SOA mass was analyzed.

Response: We thank the reviewer for pointing this out. A new figure showing the particle size distribution is given in Figure SI1. The SOA mass collected on the filters is given on page 3 line 25 of the revised manuscript.

The author concluded in page 6, lines 20-23 that ROOH do not contribute significantly to the dimer and trimer SOA signals. Then in page 3, line 32, the author described: "vaporizer temperature and ion transfer tubing temperature are set at 200 C."Previous studies have shown that oligomers from alpha-pinene SOA can thermally decompose at 100 C (Hall and Johnston 2012, Williams, Zhang et al., 2016). Because the author heated it to 200 C, would be possible that a lot of oligomers are decomposed to form small ROOH molecules when passing through the vaporizer and the ion transfer tubing, leading to a bias of the current results?

Response: We agree with the reviewer that the high vaporizer and ion transfer tubing temperatures may lead to decomposition of some of the SOA dimers and trimers, and we now make this point in the paper (page 6 line32-33). We note that when we analysed ROOH standards (tert-butyl hydroperoxide, cumene hydroperoxide, 2-butanone peroxide, and ISOPOOH), the effects of the vaporizer and ion transfer tubing temperatures on the ROOH signals were investigated. We found that while the ROOH signals at 100 °C were much lower than those at 200 °C and 300 °C, a significant fragmentation of the ROOH was observed at 300 °C. As a result, we set the vaporizer and ion transfer tubing temperatures at 200 °C throughout the experiments. Of course, we do not know if this temperature does not lead to decomposition of dimeric ROOH species.

References

Hall, W. A. and M. V. Johnston (2012). "The Thermal-Stability of Oligomers in Alpha-pinene Secondary Organic Aerosol." Aerosol Sci. Technol. 46(9): 983-989. Williams, B. J., et al. (2016). "Organic and Inorganic Decomposition Products from the Thermal Desorption of Atmospheric Particles." Atmos. Meas. Tech. 9(4): 1569-1586.

---

## Author Comment (AC3) · 16 Apr 2018

Zhou et al. propose a new method to detect the hydroperoxides and peroxy acids in complex matrices such as organic aerosol. This analytical method is based on atmospheric pressure chemical ionization coupled with a mass spectrometer. While this study presents a promising method, additional discussions/information are needed before the publication.

[Figure]

General comments: What is the purity of the standards synthesized? It is important to know the presence of potential artifacts to better understand the MS data. NMR data would be appreciated.

Response: We reported the purities of the ROOH and other standards in the manuscript on page 3 line 1-5. As none of the ROOH standards are 100% pure, this makes the interpretation of the full scan and SIM mode mass spectra of ROOH difficult. We mentioned this point in the paper on page 4 line 23-26: "Note that we do not attempt to interpret the mass spectra of ROOH and other standards obtained under full scan and SIM modes due to the presence of stabilizers and other impurities in the standard samples that make the mass spectra complex. Instead, we focus on the protonated and ammoniated molecular ions of the ROOH molecules." In particular, we do not feel that NMR data would enhance the paper because mass spectral data are only reported from the ROOH components of the samples, and not from the impurities.

Page 3. Lines 31-32: As the authors mentioned the hydroperoxides are thermally unstable, so did they try to optimize the temperature of the vaporizer? Does it have an effect?

Response: This question has been addressed above in our response to Reviewer 2.

Page 4. Lines 23-26: it is not clear why the authors decided to not look at the MS into more details. I can understand that in the MS without the addition of AA it is not necessarily worth it but in the presence of AA I do think it is important to have a closer look to check if other fragmentations (specific?) occurred. So the authors should dig into the MS to look at the presence of other specific fragments or clusters.

Response: As mentioned above and in the paper, the purities of the ROOH standards range between 32-80%. Therefore, without knowing the composition of stabilizers and other impurities in the ROOH standards, we do not think a full interpretation of the mass spectra of ROOH is possible nor necessary, i.e. our goal is not to analyze the full composition of the commercial samples and, instead, we focus specifically on the

tandem mass spectrometry of the ionized ROOH compounds.

Page 5. Line 5 Please provide the product spectra and MS/MS spectra for all the standards tested. It is important to have such information.

Response: The product spectra for all the [M+NH4]+ standards tested (namely tert-butyl-hydroperixe, 2-butanone peroxide, peracetic acid, benzoyl peroxide, di(dodecanoyl)peroxide, di-tert-butyl peroxide, erythritol, pinonic acid, and 2-nonenal) have been added into the revised manuscript into the Figure SI7 and SI8. As can be seen from these MS/MS spectra, only the ROOH standards produce fragments with 51 Da neutral loss.

Page 5. Lines 7-16: The authors need to provide more information on the optimization of their method. It seems that the method leads to important fragmentations of the molecular ions and the fingerprint fragment ions have generally a low intensity. Have they tried to maximize the neutral loss of 51? If the authors haven't tried to optimize this aspect, what was the reason?

Response: We optimized the CID conditions, source fragmentation voltages, and ion transfer tubing voltages, as well as the sheath gas, auxiliary gas, and sweep gas flows in order to maximize the signal from the 51 Da neutral loss products. We added this point on page 4 line 7-8 and line 11-13 of the revised manuscript.

Page 5 line 25. Similar to my previous comment, the authors need to present as much data as they can in order to demonstrate the robustness of the technique. In addition, some of the MS are quite noisy and lead to the question of the sensitivity of this analytical method. While it is promising and it seems very selective for the analyses of hydroperoxides, the results presented here indicate that it is not a sensitive method. What is the SOA mass collected from the chamber experiments? The LOD reported page 6 are very bad ($\sim$mg L-1) compare to the current standards in the characterization of organic aerosol ($\sim$ng L-1). The authors should discuss this issue especially in the perspective of the chemical characterization of hydroperoxides within environmental matrices (e.g. SOA) and the instability of such products. Page 7. line 10-12 The authors claim that MS/MS will yield low detection limit and allow the identification of the hydroperoxides & peroxy acids. Based on the MS/MS spectra presented and the concentrations used in this study I don't think the authors can claim to propose a method with "a low detection limit". In addition, the authors need to report in the captions the concentrations of the solution and/or the mass of organics needed to generate the MS data.

Response: We thank the reviewer for making these points. We agree that the neutral loss of 51 Da from the ammoniated molecular ions of the ROOH standards may not lead to the most intense fragments in the ROOH product spectra. This will compromise the sensitivity of the method. Nevertheless, the purpose of this work is to illustrate a new, selective analytical method which is characteristic for ROOH products that can be used to identify/quantify ROOH in complex matrices. To specifically address the point about limit of detection (LOD), we performed additional experiments to determine the LOD of cumene hydroperoxide by applying selective reaction monitoring (SRM, m/z 170>m/z 119) in the LC-MS/MS analysis. The chromatogram and the calibration curve are presented in Figure SI2. As expected, the LOD is much lower: i.e. 0.9 $\mu$M, more than 300 times lower than with the direct infusion experiments. This confirms our expectation that MS/MS analysis coupled to liquid chromatography can lower the LOD in ROOH analysis. We added this argument on page 5 line 23-24.

Page 7. Lines 14-16 The authors should propose similar Figure as Figure 5 for the SOA generated under humid conditions. The authors have to be prudent in their conclusions and keep in mind the high reactivity of oligomeric products formed from the reaction of Criegee radicals with protic substances (e.g. Riva et al., 2017 Atmos Environ). Indeed, such species can have been degraded during the 72 hours of sampling and/or throughout the analytical protocol. Did the authors estimate the degradation of the hydroperoxides/peracid?

Response: As we mentioned in the manuscript, the mass spectrum of SOA formed

under humid conditions (RH=50%) is extremely similar to that under dry conditions (RH<5%) and so we do not think it is necessary to present it in the paper, but it is in the SI (Figure SI11). We added the following sentences on page 6 line 33 to page 7 line 2 of the revised manuscript to address the reviewer's comment: "As well, while the ROOH formed under dry and humid conditions show similar mass spectra (Figures SI11), we note that more $\alpha$-hydroxy hydroperoxides may be formed under humid conditions as a result of Criegee reactions with water, which may decompose during the ionization processes due to their thermal instability". This might be the cause of the similar mass spectral patterns.

Technical comments: m/z should be in italic.

Response: This has been changed.

Page 6. Proper references for the formation of dimers: Crounse et al 2013 and Ehn et al. 2014.

Response: These two references have been cited in the revised manuscript.